# Regional Flood Frequency Analysis Using the FCM-ANFIS Algorithm: A Case Study in South-Eastern Australia

Amir Zalnezhad [1], Ataur Rahman [1,*], Mehdi Vafakhah [2], Bijan Samali [1] and Farhad Ahamed [3]

1   School of Engineering, Design and Built Environment, Western Sydney University, Penrith, NSW 2751, Australia; a.zalnezhad@westernsydney.edu.au (A.Z.); b.samali@westernsydney.edu.au (B.S.)
2   Department of Watershed Management, Faculty of Natural Resources, Tarbiat Modares University, Noor P.O. Box 46417–76489, Mazandaran Province, Iran; vafakhah@modares.ac.ir
3   School of Computer, Data and Mathematical Sciences, Western Sydney University, Penrith, NSW 2751, Australia; farhad.ahamed@westernsydney.edu.au
*   Correspondence: a.rahman@westernsydney.edu.au

**Abstract:** Regional flood frequency analysis (RFFA) is widely used to estimate design floods in ungauged catchments. Both linear and non-linear methods are adopted in RFFA. The development of the non-linear RFFA method Adaptive Neuro-fuzzy Inference System (ANFIS) using data from 181 gauged catchments in south-eastern Australia is presented in this study. Three different types of ANFIS models, Fuzzy C-mean (FCM), Subtractive Clustering (SC), and Grid Partitioning (GP) were adopted, and the results were compared with the Quantile Regression Technique (QRT). It was found that FCM performs better (with relative error (RE) values in the range of 38–60%) than the SC (RE of 44–69%) and GP (RE of 42–78%) models. The FCM performs better for smaller to medium ARIs (2 to 20 years) (ARI of five years having the best performance), and in New South Wales, over Victoria. In many aspects, the QRT and FCM models perform very similarly. These developed RFFA models can be used in south-eastern Australia to derive more accurate flood quantiles. The developed method can easily be adapted to other parts of Australia and other countries. The results of this study will assist in updating the Australian Rainfall Runoff (national guide)-recommended RFFA technique.

**Keywords:** ANFIS; flood; fuzzy logic; RFFA; ungauged catchments; Australian Rainfall and Runoff

## 1. Introduction

Floods often cause extensive damage and disruption to human life. Losses of life, social impacts, homelessness, and financial loss such as property and infrastructure damage are the most common impacts of large flood events. In recent years, devastating floods in Europe, China, Australia, Iran, and other parts of the world have been linked to climate change [1].

Hydrologists use a risk-based method in flood estimation; in this regard, the term 'design flood' is often used, which is defined as flood discharge associated with an annual exceedance probability (AEP) or average recurrence interval (ARI) or return period. For gauged catchments, at-site flood frequency analysis (FFA) is generally adopted to estimate design floods. For ungauged catchments or for sites with poor quality/inadequate streamflow data, regional flood frequency analysis (RFFA) is used. There have been numerous research studies on both at-site FFA and RFFA around the globe, such as GIS-based methods [2], region-of-influence (ROI)-based RFFA methods [3,4], RFFA using wavelet neural networks [5], index flood methods (IFM) [6,7], regression-based RFFA methods [8,9], artificial intelligence (AI)-based methods [10,11], random forest forming regions in RFFA [12,13], L-moments based index flood methods [14,15], and Rational Method [16].

During the past two decades, many different forms of linear RFFA techniques have been proposed such as IFM in south-east Australia [17]; USA [18]; UK [19]; in New South

Wales (NSW), Australia [20]; in Canada [21]; and in Czech Republic [22] and Quantile Regression Technique (QRT) in northern USA [23]; UK [24]; NSW [25]; and USA [26]. These linear techniques cannot account for the inherent non-linearity in the rainfall-runoff process, and hence several non-linear RFFA techniques have been proposed, including Adaptive Neuro-Fuzzy Inference Systems (ANFIS) [27,28], Artificial Neural Network (ANN) [29,30], Support Vector Machine (SVM) [31,32], and Machine Learning Models (MLMS) [33].

ANFIS has been used to identify non-linear relationships between chaotic data [27]. For example, Bozchaloei and Vafakhah [34] applied ANN and ANFIS methods to study flow duration curves at Namak Lake catchment in Iran using 21 years of data from 33 catchments and found that ANFIS had a better performance than other methods. Riahi-Madvar et al. [35] studied streamflow prediction at Kuran River in Iran, where they used different variations of ANFIS mixed with other methods such as Fire-Fly Algorithm (FFA), Genetic Algorithm (GA), Grey Wolf Optimization (GWO), Particle Swarm Optimization (PSO), and Differential Evolution (DE) hybridized and found that the ANFIS-GWO outperformed other techniques. Sharifi Garmdareh et al. [36] used ANFIS, ANN, and SVM models in RFFA at Namak Lake using data from 55 hydrometric stations with 20 years of flood discharge data, and found that ANFIS and SVM models outperformed the ANN model.

Shu and Ouarda [37] studied the performance of the ANFIS model in Quebec, Canada, using data from 151 gauging stations, and found ANFIS as a useful method in estimating flood quantiles. Mukerji et al. [38] compared three models, including ANFIS, to predict design floods at the Ajay River Basin in Jharkhand, India. They compared the relative performance of ANN, ANGIS, and ANFIS techniques based on 20 rainfall-runoff events. They demonstrated that the ANGIS model had the best performance, and in most cases, the ANFIS model outperformed the ANN model. Aziz et al. [39] used data from 452 catchments to compare the performance of linear and non-linear RFFA techniques in south-east Australia. In their study, the ANN was found to generate more accurate flood quantiles than the Co-Active Neuro Fuzzy Inference System (CANFIS). Jimeno-Sáez et al. [40] applied ANN and ANFIS to estimate peak flows using over 30 years of recorded flow data from 14 gauging stations in Spain. They found that the ANFIS model outperformed the ANN model. Different variations and combinations of artificial intelligence (AI)-based models were adopted in RFFA by many other researchers [41,42].

In some of these comparison studies, it can be seen that AI-based RFFA models performed better than the common linear RFFA techniques. In this study, ANFIS is selected over other AI-based methods as it is one of most widely used techniques in hydrology [10,35] and performed better than other AI-based techniques as noted above. Comparing the ANFIS model with the QRT model for eastern parts of Australia seems to be necessary since there are very limited studies on the application of non-linear RFFA techniques in Australia and internationally. The motivation of this study is to create sound scientific basis on the ANFIS-based RFFA techniques based on the recent flood data, which will assist in the upgradation of the RFFA techniques in the national Australian Rainfall and Runoff guide. The objectives of this study are: (i) to develop and test three different non-linear/AI-based RFFA techniques; (ii) to identify the best performing AI-based RFFA technique in south-east Australia; and (iii) to compare non-linear AI-based RFFA techniques with QRT (linear method).

## 2. Material and Methods

### 2.1. Study Area

Southeast Australia (NSW and Victoria (VIC) states) was selected for this study since this area of Australia has the best quality hydrological data. This region is home to about 57% of the country's population

A total of 181 catchments were selected from the study area. The selected catchments are natural and have not been affected by any major land-use changes. Figure 1 shows the locations of selected catchments (training and test catchments). The annual maximum flow

record length of these 181 catchments varies from 40 to 89 years (average = 48 years). Flood quantiles for AEP of 1 in 2, 1 in 5, 1 in 10, 1 on 20, 1 in 50, and 1 in 100 years ($Q_2$, $Q_5$, $Q_{10}$, $Q_{20}$, $Q_{50}$, and $Q_{100}$, respectively) were estimated using an LP3 distribution for each of the selected 181 catchments. A Bayesian fitting procedure based on the method of moments was adopted to fit the log-Pearson Type 3 (LP3) distribution [43]. As shown in Table 1, a total of eight catchment characteristics were selected as predictor variables. These were selected as previous studies in this region found these to be the most important predictor variables in RFFA [3,25]. Using a random selection method, 126 catchments were selected as a training data set, leaving 55 as the testing data set.

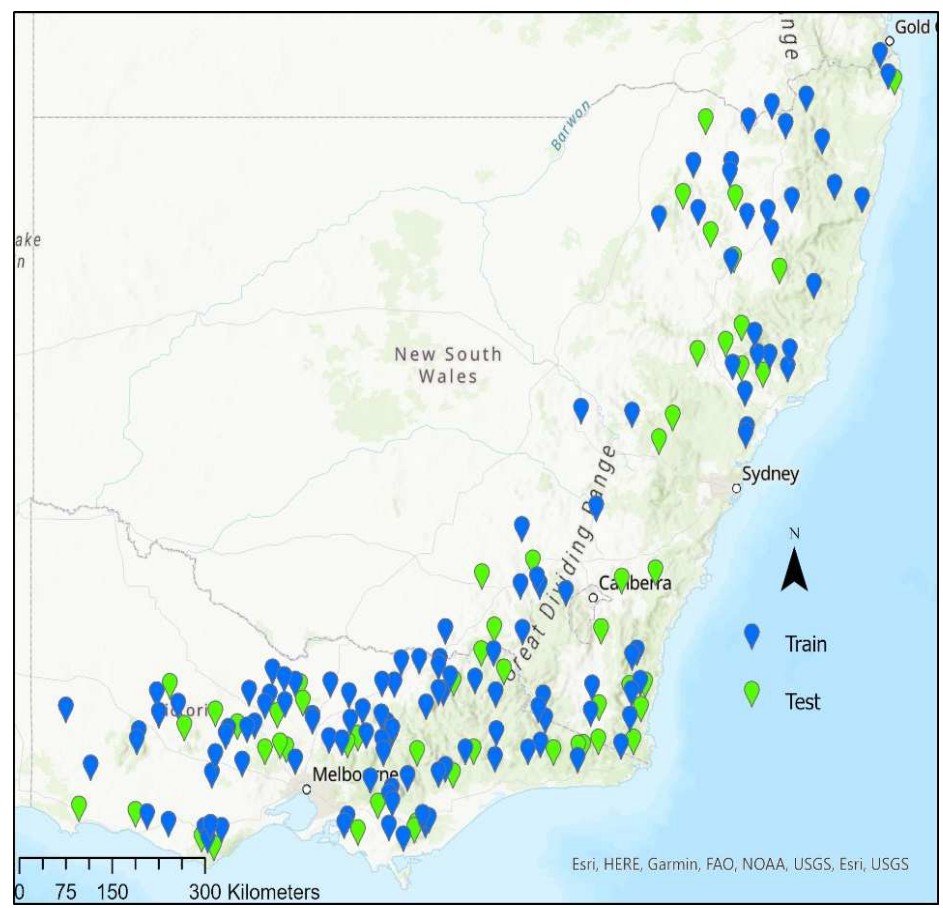

**Figure 1.** Location of the selected catchments.

**Table 1.** Important statistics of the selected predictor variables for 181 study catchments.

| Predictor Variable | Name of Variable | Unit | Statistical Parameter | | |
|---|---|---|---|---|---|
| | | | Minimum | Maximum | Mean |
| AREA | Catchment area | km$^2$ | 3.00 | 1010.00 | 349.06 |
| $I_{62}$ | Design rainfall intensity with 6 h duration and 2 years return period | mm/h | 24.60 | 87.30 | 39.03 |
| MAR | Mean annual rainfall | mm | 484.39 | 1953.23 | 970.50 |
| SF | Shape factor | - | 0.25 | 1.62 | 0.78 |
| MAE | Mean annual evapo-transpiration | mm | 925.90 | 1543.30 | 1112.74 |
| SDEN | Stream density | km$^{-1}$ | 0.51 | 5.47 | 2.06 |
| S1085 | Slope of central 75% of the mainstream | m/km | 0.80 | 69.90 | 13.02 |
| FOREST | Fraction forest | - | 0.00 | 1.00 | 0.55 |

### 2.1.1. Catchment Area (AREA)

Catchment area (AREA) is one of the main morphometric characteristics of a catchment, which impacts many other characteristics. The AREA of a catchment usually directly relates to other characteristics such as stream length and slope of a catchment [7,44]. This predictor variable has been used in almost all the RFFA studies and is regarded as a scaling factor in the rainfall-runoff process. It was measured using a planimeter on the topographic map (1:100,000 scale).

### 2.1.2. Design Rainfall Intensity (I62)

Rainfall intensity directly impacts flood generation processes and has been used in many RFFA studies [3,9]. To define design rainfall intensity, one needs to select a rainfall duration and return period. In this study, a duration of 6 hours with a return period of 2 years was selected to obtain design rainfall intensity at the catchment centroid, following the approach of ARR [9]. This was obtained from the Australian Bureau of Meteorology's (BOM) website using the intensity-frequency-duration (IFD) tool.

### 2.1.3. Mean Annual Rainfall (MAR)

MAR does not affect flood peaks directly, but it is a surrogate to many other catchment characteristics. It was obtained from the BOM website.

### 2.1.4. Shape Factor (SF)

The shape of a catchment directly impacts the flood generation process, e.g., flood responses are usually slower in elongated catchments than rounder ones. The shape factor was calculated by dividing the shortest distance between the catchment outlet and catchment centre by the square root of AREA [9].

### 2.1.5. Mean Annual Evapotranspiration (MAE)

MAE data was obtained from BOM website [9]. MAE, also known as the dryness index, similar to MAR, is indirectly related to the flood generation process.

### 2.1.6. Stream Density (SDEN)

SDEN is one of the most influential factors in the rainfall-runoff process. The more and longer tributaries there are in a catchment, the higher the stream density, e.g., more tributaries let water move into the mainstream faster, resulting in a larger peak flow. It was calculated by dividing the total stream length within the catchment (on a 1:100,000 topographic map) by AREA.

### 2.1.7. S1085

The mainstream slope (S1085) affects water velocity. This factor was measured based on the mid 75% of the stream length, excluding the bottom 10% and top 15% of the catchment length. The higher the S1085, the quicker the flood response. S1085 was calculated using the following equation:

$$S1085 = \frac{(H2 - H1)}{(0.75 \times L)} \tag{1}$$

where $H2$ and $H1$ are heights above sea level at 0.85, 0.10 points of the length of the mainstream, measured from catchment outlet, and $L$ is the mainstream length.

### 2.1.8. Forest

FOREST is a primary factor in determining water loss through infiltration during a rainfall event, e.g., the higher the FOREST, the lower the pace of flood generation, as FOREST increases the roughness of a catchment. This factor was calculated by dividing

the forested area by the total catchment area (AREA), and the area was measured using a planimeter on a topographic map with a scale of 1:100,000.

*2.2. Methodology*

2.2.1. Adaptive Neuro-Fuzzy Inference System (ANFIS)

In a fuzzy system, input and output data are defined linguistically, and in case of a lack or inconsistency of information, a fuzzy system is an excellent tool for adaptation [45]. The most popular Fuzzy Logic (FL) methods that are implemented include those that are suggested by Assilian and Mamdani [46] and Takagi and Sugeno [47]. ANFIS is a fuzzy system based on Takagi and Sugeno's method that performs as a network of nodes and sets of parameters linked to each other. As shown in Figure 2, the ANFIS model consists of 5 different layers between raw input data and model outputs. Due to the learning capability, this model uses a set of rules to change the parameters to minimize the model error [27,48]. ANFIS is based on a collection of 'if-then' fuzzy rules that gradually train the data. In this study, three common types of ANFIS, Grid Partitioning (GP), Subtractive Clustering (SC), and Fuzzy C-mean (FCM) are used, and the best results are compared with the QRT.

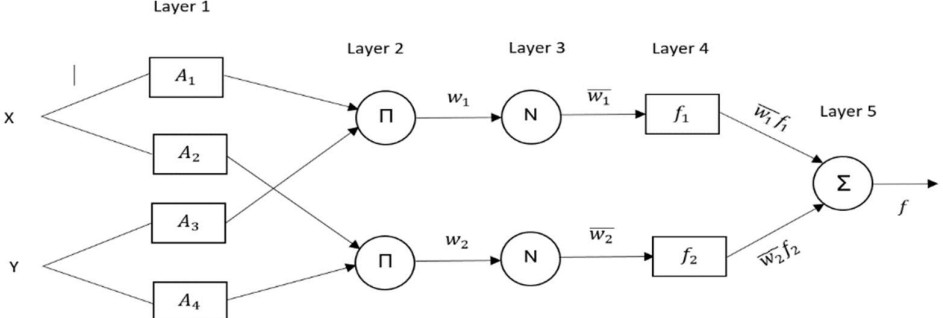

**Figure 2.** ANFIS model based on two input variables and one output.

2.2.2. Grid Partitioning (GP)

The structure of the ANFIS model consists of five layers, including premise and consequence parts, e.g., for only two inputs and one output with nine rules, as shown in Figure 2.

Layer 1: fuzzification layer

In this layer, input values are divided into fuzzy clusters using membership functions, shown in Equation (2), where $O_i^1$ is the membership function of $A_i$, and $A_i$ is a linguistic label. A set of parameters such as (a, b, c) are used in this layer to calculate the membership degrees of each function shown by $\mu(x)$. Several differentiable functions, including bell-shaped, trapezoidal, and triangular, can be used to define $\mu(x)$, e.g., the bell-shaped function as shown in Equation (3).

$$O_i^1 = \mu_{A_i}(x) \tag{2}$$

$$\mu_{A_i}(x) = \frac{1}{1 + \left[\left(\frac{x - c_i}{a_i}\right)^2\right]^{b_i}} \tag{3}$$

Layer 2: rule layer

As shown in Equation (4), incoming values of membership degrees turn into firing strength as shown with wi.

$$w_i = \mu_{A_i}(x) \times \mu_{B_i}(y), \ i = 1, \ 2 \tag{4}$$

Layer 3: normalization layer

In this layer, firing strength values produced in the rule layer are normalised using Equation (5).

$$\overline{w_i} = \frac{w_i}{w_1 + w_2} \quad i = 1, 2.$$  (5)

Layer 4: defuzzification layer

In this layer, as shown in Equation (6), weighted values for each rule are calculated using normalized firing strength values produced in layer 3.

$$\overline{w_i} f_i = \overline{w_i}(p_i x + q_i y + r_i)$$  (6)

where $p_i$, $q_i$, and $r_i$ are the parameter set also known as consequence parameters.

Layer 5: summation layer

In this layer, output of layer 4 is summed by using Equation (7) to produce the final result for ANFIS.

$$ANFIS \ output = \sum_i \overline{w_i} f_i = \frac{\sum_i w_i f_i}{\sum_i w_i}$$  (7)

In this study, the MATLAB code was used for training and testing the *ANFIS* model. This code uses the maximum, product, summation, and the weighted average for And, Or, Implication, Aggregation, and Defuzzification methods for Grid partitioning-ANFIS models.

### 2.2.3. Subtractive Clustering (SC)

This method is a partitioning method that determines cluster centres based on the density measurement of each data point. Initially, all data has the potential to become a data centre. However, after normalizing and scaling the data, the density of each data is defined according to Equation (8).

$$D_i = \sum_{j=1}^{N} \exp\left(-\left(\frac{2}{r_a}\right)^2 . ||x_i - x_j||^2\right)$$  (8)

where $D_i$ is a density measure, $r_a$ is the influence radius, $x_i$ is the regarded cluster centre, and $x_j$ is the remaining data point. Since data with adjacent data are more likely to be at the centre of a cluster, they are first selected as the cluster's centre. Then, in the successive iterations, the new centre points are calculated according to Equation (9) using the centre point of the former iteration.

$$D_i = D_i - D_{ci} \sum_{j=1}^{N} \exp\left(-\left(\frac{2}{r_a}\right)^2 . ||x_i - x_j||^2\right)$$  (9)

where $D_{ci}$ is the centre point of the previous iteration. In each iteration, the centre point is defined as the highest density point in the corresponding cluster. Once a new central point is identified, the density of the data points is revised, and this process continues until an appropriate number of clusters are generated. Therefore, the final identified centre for the clusters is used for the fuzzy rules. Chiu [49] and Chiu [50] have also suggested other automatically-identified clustering methods for SC.

### 2.2.4. Fuzzy C-Means (FCM)

The FCM model reduces the non-linearity of data by grouping the training data based on similar attributions. For classification purposes, this method assigns each data to the corresponding category based on its distance to the centre of the category. The FCM algorithm was suggested by [51]. In this method, each point of a data set belongs to different groups using membership functions, which ranges between zero and one, so that the sum of their total (membership functions) for each data point is one. In the FCM method, $n$ data points are divided into c fuzzy clusters ($c < n$), and the location of these

clusters would be appropriately adjusted. The FCM model is based on the square error distance, and this concept is considered as the following mathematical basis.

In the following equations, $x_k$ is assumed as the $k'$th vector data-point ($k = 1, 2, \ldots, n$). $v_i$ is the centre of the $i'$th cluster ($I = 1, 2, \ldots, C$). Error distance $d_{ik}$ of $x_k$ is calculated as follows: $d_{ik} = || \, x_k - v_k \, ||$. In Equation (10), the degree of membership of $k'$th data-point in the cluster $i$ is denoted by $u_{ik}$ [52], where:

$$\sum_{i=1}^{C} u_{ik} = 1 \tag{10}$$

Furthermore, the purpose is to minimize the following function [52]:

$$J(U, v) = \sum_{i=1}^{C} \sum_{k=1}^{n} (u_{ik})^m (d_{ik})^2 \tag{11}$$

The exponent parameter, $m$, should satisfy $1 < m < \infty$, and is generally chosen to be $m = 2$ [53]. It is also called the fuzzifier parameter and adjusts the fuzziness of clustering [54,55], e.g., in the larger amounts of $m$ ($m \rightarrow \infty$), the membership degrees of data-points tend to be well-fuzzy [52].

The FCM algorithm could be described in the following steps, as introduced by Nourani and Komasi [53]:

1.  Assuming initial locations for each cluster centre.
2.  Each data-point joins the cluster with the nearest cluster centre.
3.  New cluster centres are computed and considered the centroids of clusters.
4.  Terminate the process if the cluster partition is stable. If not, go to the second step.

For step 2, the membership value can be expressed by the Lagrange multiplier technique, Equation (12) [52]:

$$u_{ik} = 1 / \left\{ \sum_{j=1}^{n} \left[ d_{ik}^2 \Big/ d_{jk}^2 \right]^{\frac{1}{m-1}} \right\} \tag{12}$$

After the membership degree of data-points to clusters are crisp, one gets [53]:

$$u_{ik} = 0; \, \forall \, i \neq j$$

$$u_{jk} = 1; j \, s.t. \, d_{jk} = \min\left( d_{jk}, \, i = 1, 2, \ldots, c \right)$$

Cluster centres at the third step can be recalculated via Equation (13) [52]:

$$v_i = \frac{\sum_{k=1}^{n} (u_{ik})^m x_k}{\sum_{k=1}^{n} (u_{ik})^m} \, i = 1, 2, \ldots, c \tag{13}$$

where $u_{ik}$ and $v_i$ are calculated by minimizing the J(U,v) using the following Lagrangian objective function:

$$min\{ \, J_m(U, V, \lambda; x) = \sum_{i=1}^{c} \sum_{j=1}^{n} u_{ij}^m || \, x_i - v_i ||^2 + \\ \sum_{k=1}^{n} \lambda_k \sum_{i=1}^{c} (u_{ik} - 1) \} \tag{14}$$

$$\text{and computing} \left( \frac{dJ_m}{dv_i} = 0 \, \frac{dJ_m}{du_{ij}} = 0 \, \frac{dJ_m}{d\lambda_k} = 0 \right)$$

In this study, the Darbin–Watson statistic is used to examine the degree of correlation among the predictor variables.

### 2.2.5. Quantile Regression Technique (QRT)

QRT, proposed by the United States Geological Survey (USGS), is a regression-based method that uses streamflow, catchment characteristics, and climatic variables data to

develop a prediction equation for an individual return period of interest [18,26]. This method can be expressed by Equation (15):

$$log(Q_T) = a + b \times log(B) + c \times log(C) + d \times log(D) + \ldots \tag{15}$$

where $Q_T$ = flood quantile for ARI of $T$ years; $a, b, c, d, \ldots$ are regression coefficients; and $B$, $C, D, \ldots$ are catchment characteristics (predictors).

In this study, the ordinary least squares (OLS) and a backward variable selection method were adopted in SPSS software to estimate the regression coefficients. Final regression equations were developed based on predictor variables with a *p*-value of less than or equal to 0.10.

2.2.6. Statistical Measures Used in Evaluation

Based on the model prediction (Qpred) for the 55 test catchments, the following evaluation statistics were used to compare the performance of different RFFA techniques.

$$RE = \frac{Qpred - Qobs}{Qobs} \times 100 \tag{16}$$

$$REr = median[abs(RE)] \tag{17}$$

$$RMSE = \sqrt{mean\left[(Qpred - Qobs)^2\right]} \tag{18}$$

$$RBias = \left[mean\left(\frac{Qpred - Qobs}{Qobs}\right)\right] \times 100 \tag{19}$$

$$RRMSE = \frac{\sqrt{mean\left[(Qpred - Qobs)^2\right]}}{mean(Qobs)} \tag{20}$$

$$RMSNE = \sqrt{mean\left[\left(\frac{Qpred - Qobs}{Qobs}\right)^2\right]} \tag{21}$$

*Qobs* is an at-site flood quantile obtained by at-site FFA (using LP3 distribution), and *Qpred* is predicted-flood quantiles using either ANFIS or QRT for each test catchment.

**3. Results and Discussion**

Three types of ANFIS models, FCM, SC, and GP, were developed. For the FCM model, the training data set was divided into 2–50 clusters to find the best possible number of clusters; e.g., for the $Q_2$ model, data were divided into eight clusters (due to large errors, models with the number of clusters less than ten are shown in Table 2). For SC, different radius values were tested, ranging from 0.30–0.99, e.g., for $Q_2$ model rad = 0.76 was selected since it produced the best RMSE value. For GP, seven different membership function types were tested, including trimf, trapmf, gaussmf, gbellmf, gauss2mf, dsigmf, and pimf, and membership function numbers ranging from two to five, e.g., triangular membership function type with membership function number of two was found to be the best performing model for $Q_2$.

The best performing parameters based on the RMSE criteria for the three types of ANFIS model for $Q_2$–$Q_{100}$ are shown in Table 2. For the FCM model, it was found that dividing data into two to eight clusters produced relatively better results (Table 2). For the SC model, the best performing rad was found to be between 0.76–0.79 for $Q_2$, $Q_{20}$, $Q_{50}$ and $Q_{100}$, and 0.39 and 0.37 for $Q_5$ and $Q_{10}$, respectively. For the GP model, the triangular membership function type with membership function number = two, was found to be the best performing parameter for $Q_2$, $Q_5$, $Q_{10}$, and $Q_{100}$. However, for $Q_{20}$ and $Q_{50}$, the gauss2mf model exhibited better performance.

**Table 2.** Rad_SC, nCluster _FCM, mf-type, and mf_n for GP.

|  | nCluster_FCM | Rad_SC | MF-Type and mf_n_GP |
|---|---|---|---|
| $Q_2$ | 8 | 0.76 | Trimf-2 |
| $Q_5$ | 2 | 0.39 | Trimf-2 |
| $Q_{10}$ | 5 | 0.37 | Trimf-2 |
| $Q_{20}$ | 3 | 0.77 | Gauss2mf-5 |
| $Q_{50}$ | 2 | 0.78 | Gauss2mf-5 |
| $Q_{100}$ | 3 | 0.79 | Trimf-2 |

The top-seven of the 35 best-performing membership functions and numbers for each of the quantiles are shown in Table 3, and this categorization is based on RMSE. The Trimf-2 model appears to be one of the seven best-performing membership functions.

**Table 3.** Seven best-membership functions based on RMSE.

| $Q_2$ | $Q_5$ | $Q_{10}$ | $Q_{20}$ | $Q_{50}$ | $Q_{100}$ |
|---|---|---|---|---|---|
| Trimf-2 | Trimf-2 | Trimf-2 | Gauss2mf-5 | Gauss2mf-5 | Trimf-2 |
| Gauss2mf-5 | Trapmf-2 | Pimf-2 | Dsigmf-5 | Dsigmf-5 | Gauss2mf-5 |
| dsigmf-5 | Pimf-2 | Trapmf-2 | Trapmf-5 | Trapmf-5 | Trapmf-5 |
| Trapmf-5 | Gaussmf-2 | Dsigmf-2 | Trimf-2 | Gauss2mf-2 | Dsigmf-5 |
| Gbellmf-5 | Gbellmf-2 | Gauss2mf-2 | Gaussmf-2 | Trimf-2 | Pimf-5 |
| Gaussmf-2 | Gauss2mf-2 | Gbellmf-2 | Trapmf-2 | Gbellmf-5 | Gauss2mf-4 |
| Gauss2mf-2 | Dsigmf-2 | Gaussmf-2 | Gbellmf-2 | Gaussmf-5 | Trapmf-4 |

Figure 3 shows a boxplot of RE values for three different types of ANFIS models. The thick line within a box represents the median RE value. For the FCM model, RE values gradually increase with increasing return periods, with an overestimation for $Q_{10}$, $Q_{20}$, $Q_{50}$, and $Q_{100}$, with the best performance for $Q_5$, where the boxplot is the smallest in size and the median RE value is near the 0.00 line. For the SC model, an overestimation is found for $Q_{100}$; however, in mid-range return periods, e.g., $Q_{10}$, $Q_{20}$, and $Q_{50}$, the median RE values are near the 0.00 line, indicating a smaller bias. This model has the best performance for $Q_{10}$, with the smallest box width and the median RE value sitting near 0.00. For the GP model, underestimation is found for all the six quantiles; however, it performs the best for $Q_5$ as far as the width of the boxplot is concerned.

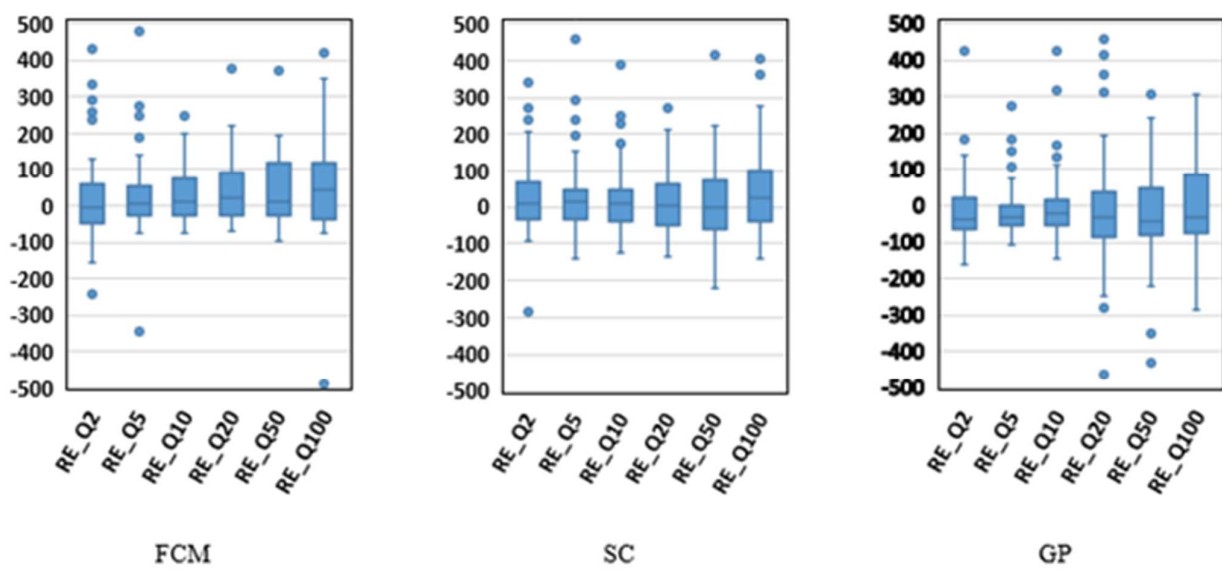

**Figure 3.** Boxplot of RE values for FCM, SC, and GP-based RFFA models.

Figure 4 shows boxplots of $Q_{\text{pred}}/Q_{\text{obs}}$ (Q_Ratio) values for FCM, SC, and GP models. As shown in this figure, a notable underestimation occurs for GP, in particular for $Q_5$ and $Q_{10}$. In terms of bias, the SC model performs the best, as in this case, the median line is

closer to the 1-1 reference line. As can be seen from Figure 4, all the models perform best for $Q_5$, having the smallest box width.

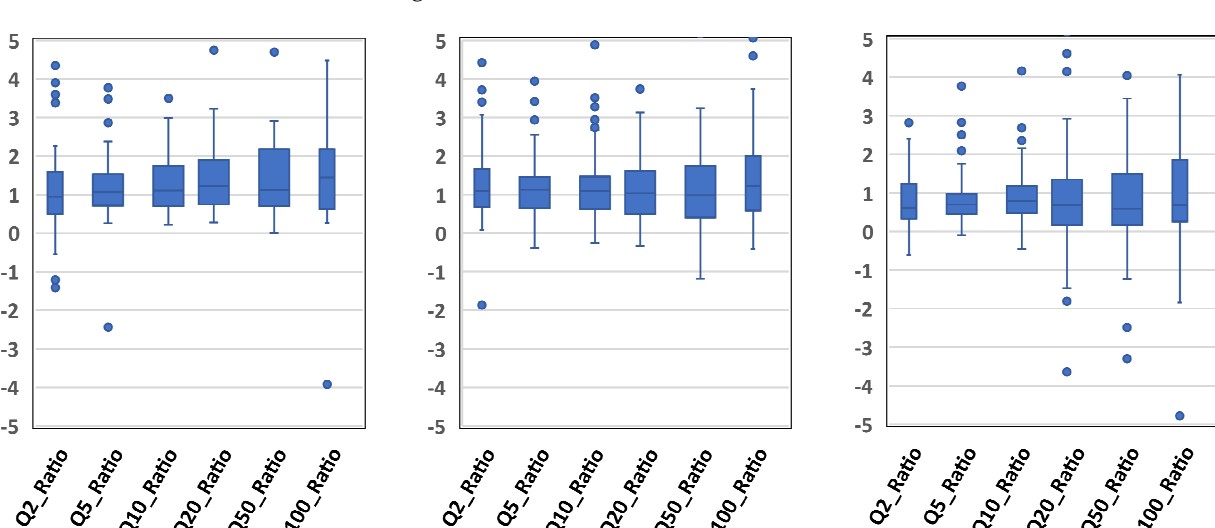

**Figure 4.** Boxplots of *Qpred/Qobs ($Q_T$_Ratio)* values for FCM, SC, and GP models.

Some important evaluation statistics such as REr, $Q_{T-\text{pred}}/Q_{T\_\text{obs}}$ (median), RRMSE, RBias, and RMSNE were adopted for validation of FCM, SC, and GP, as shown in Table S1 [56]. As seen in Table S1 (in the supplementary section), in terms of REr, the FCM model performs better in all ranges of ARIs except for $Q_2$, where SC outperforms FCM. Moreover, for $Q_{100}$, REr of FCM outperforms REr of SC by 16.3%. In terms of $Q_{\_\text{Ratio}}$, FCM exhibits a better performance in lower ranges of ARIs ($Q_2$, $Q_5$, and $Q_{10}$); however, SC performs better than FCM and GP for mid and high ARIs ($Q_{20}$, $Q_{50}$ and $Q_{100}$) with the best performance of $Q_{\_\text{Ratio}} = 0.98$ for $Q_{50}$ (1.11 for FCM and 0.47 for GP).

The REr values for the FCM and SC models range between 38.18–59.80% and 38.85–69.85%, respectively. Haddad and Rahman [57] reported similar statistical measures for linear regression models, with REr ranging between 13–42%, Rahman and Rahman [58] with REr ranging between 22–37%, and Rahman et al. [25] with REr ranging between 33–44%. In terms of RRMSE, the SC model outperforms the other two models with the smallest range for RRMSE, which is a sign of better accuracy. This statistic has been used in different AI-based studies, Ouarda and Shu [59] reported RRMSE of 0.57–0.76 for four different methods, including the ANFIS model with RRMSE of 0.57–0.64. RRMSE ranges between 0.78–1.08 for FCM and 0.71–1.00 for SC and 0.82–1.30 for GP. The best and worst values for RBias are found to be −1.24 for SC_$Q_{20}$ and −916.13 for GP_$Q_{50}$, which shows a sign of gross under-estimation. RBias is another important statistic used in some studies, e.g., Desai and Ouarda [60] reported RBias for different models ranging from −27.85 to 3.4, and Shu and Ouarda [37] reported RBias for the ANFIS model between −11 to −8. The GP model had the best RMSNE value of 0.94 for $Q_2$; however, in terms of RMSNE, the SC model performs better than FCM in lower ARIs ($Q_2$, $Q_5$, and $Q_{10}$), and FCM outperforms the other models in higher ranges of ARIs ($Q_{20}$, $Q_{50}$, and $Q_{100}$).

The cumulative percentage of catchments having a given range of REr values are shown in Figure 5 for Q50. As can be seen in this figure, the FCM model generally seems to be the best performing model because, for most quantiles, there is a greater proportion of catchments with lower ranges of REr values (0–19%, 20–39%, and 40–59%) than the other two models. However, for ARIs of 5, 10, and 100 years, all the models perform similarly.

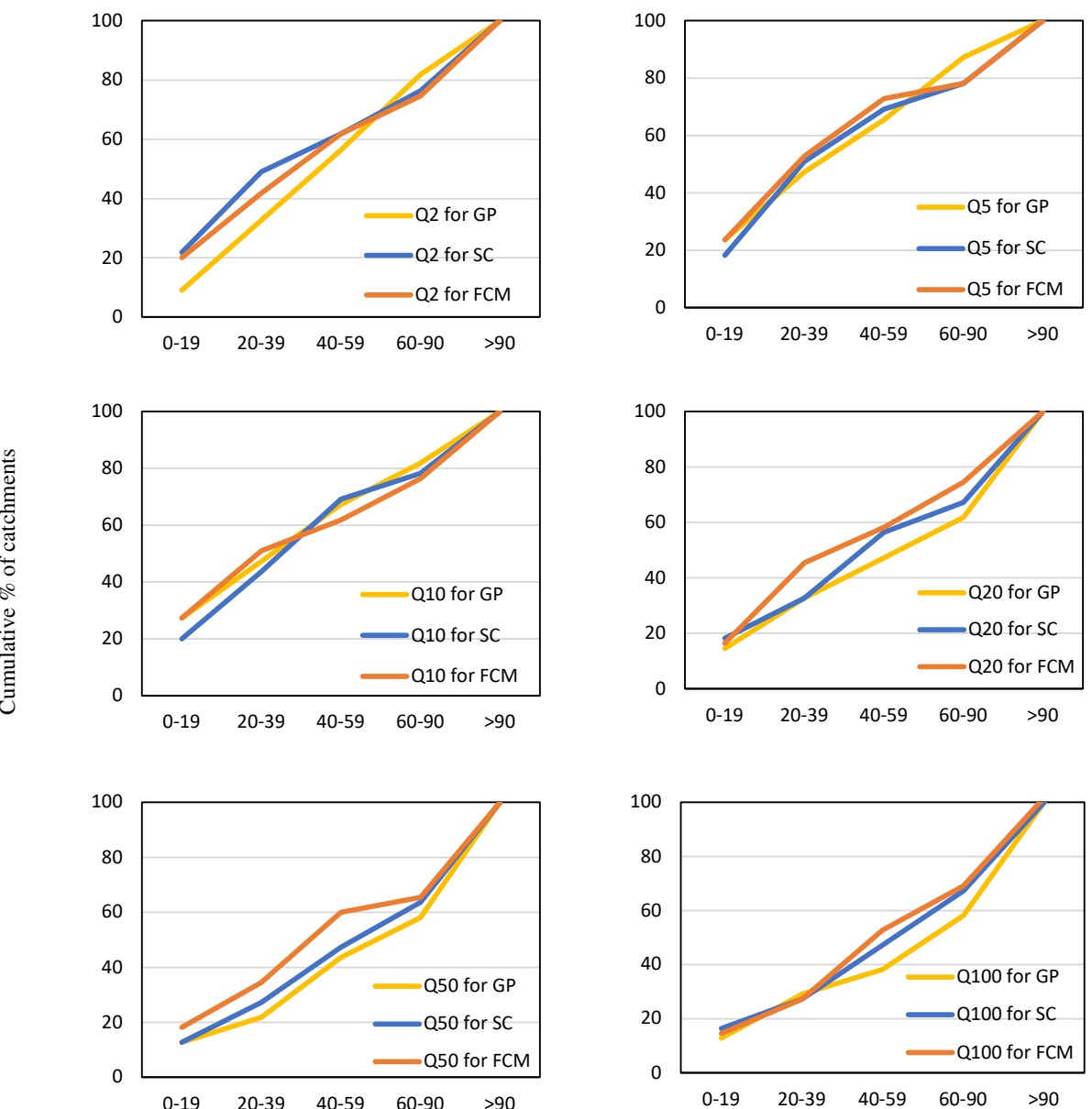

**Figure 5.** Plot of cumulative percentage of $RE_r$ for FCM, SC, and GP.

Figures 6 and 7 show qualitative comparisons of REr and Q_Ratio values between the three models (FCM, SC, and GP). Qualitative identifiers such as 'Good', 'Fair', and 'Poor' are assigned to different ranges of REr and Q_Ratio values [25]. As seen in Figure 6, 'Good' is assigned to REr values between 0 to ±30%, 'Fair' is assigned to REr values in the range of ±30% and ±60%, and 'Poor' is assigned to other stations with REr values beyond these ranges. Similarly, in Figure 7, 'Good' is a qualitative identifier for the test catchments with the Q_Ratio value between 0.8–1.3, and 'Fair' is assigned to Q_Ratio values in the range of 0.6 to 0.8, and 1.3 to 2, and 'Poor' ranking is assigned to the rest of Q_Ratio values. In terms of REr values, as shown in Figure 6, the FCM model performs better than the other models because it has more 'Good'-rated stations than the other two models. As shown in Figure 7, although both FCM and SC models outperform the GP model, the SC model seems to be a better performing model than the FCM since it has fewer 'Poor'-rated test catchments. Table S2 (in supplementary section) shows the important predictor variables for different models.

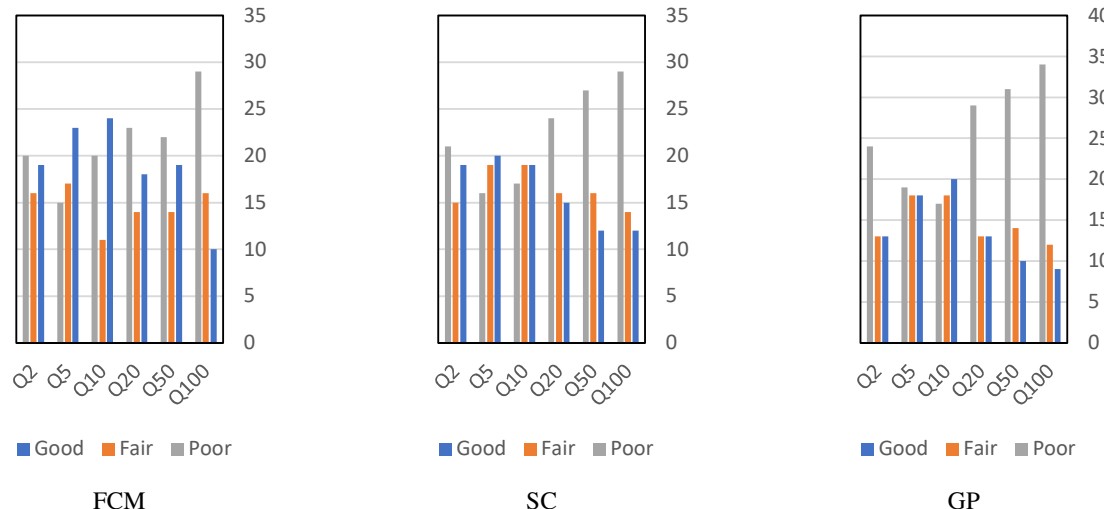

**Figure 6.** Grouping REr (%) values for FCM, SC, and GP.

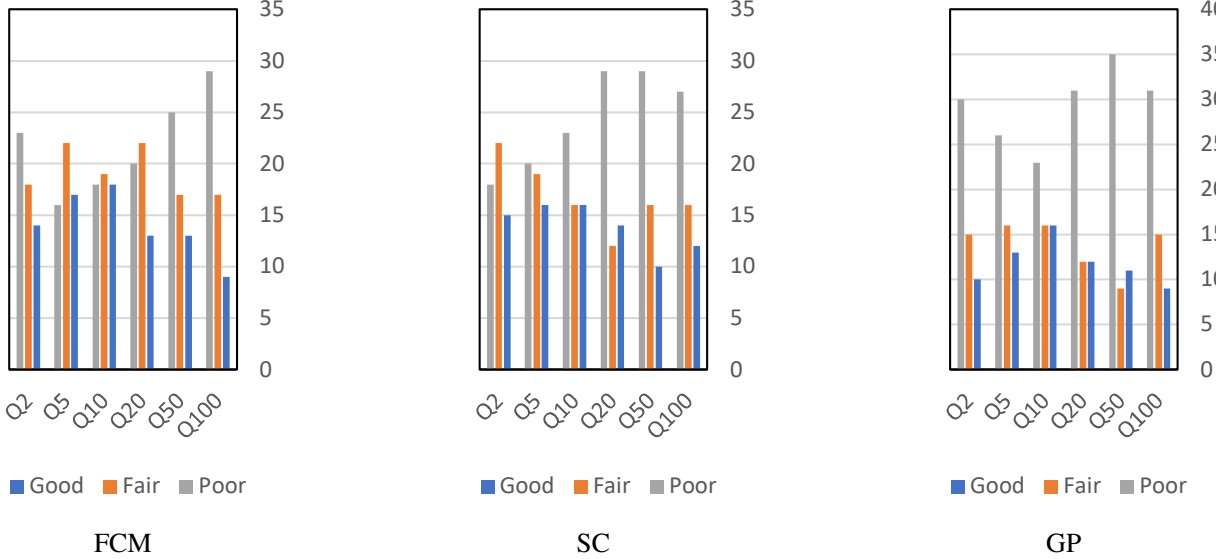

**Figure 7.** Grouping $Q_{Ratio}$ values for QRT and IFM models.

The spatial distribution of absolute REr values is presented for all the six quantiles for the FCM, SC, GP, and IFM models, e.g., $Q_{20}$ is shown in Figure 8. Green, brown, yellow, and red colours for different REr value ranges were used as visual indicators. Green triangular shape demonstrates the best performing catchments with the lowest absolute REr values of $\leq 25$. As seen from Figure 8, the FCM model performs slightly better than others for $Q_{20}$ since it has more green triangular shapes than SC and GP. For catchments close to the north-eastern and south-eastern parts of NSW, the FCM model demonstrates better performance. The SC model generally performs better in eastern NSW. Most of the poorly performing catchments are located in the southwest of NSW and VIC, with RE values of $\geq 100$, similar in all the three models. FCM shows a better performance since it has relatively fewer brown and red triangular shapes than SC and GP. Interestingly, all three models are common to most low-performance test catchments. Further investigation is needed to understand why most of the catchments located in VIC are performing poorly by ANFIS models, which is left for future research efforts.

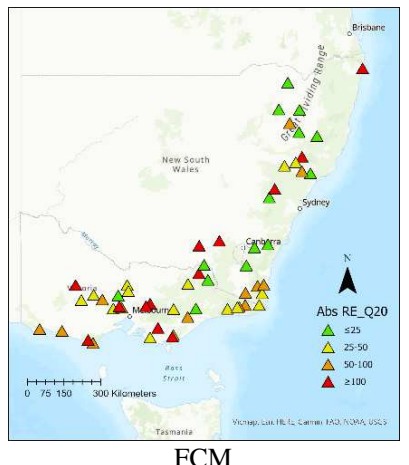 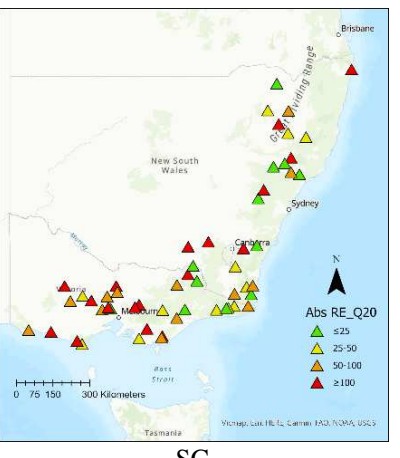 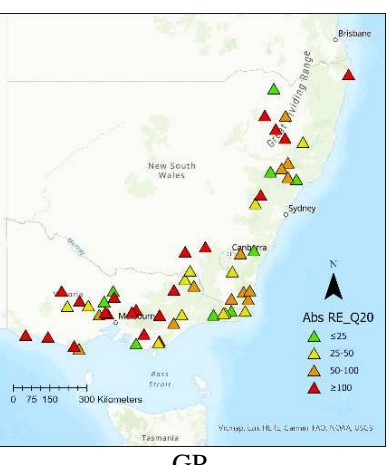

FCM　　　　　　　　　　　SC　　　　　　　　　　　GP

**Figure 8.** Spatial distribution of REr values for Q20 for FCM, SC, and GP models.

Table 4 represents the evaluation statistics for the QRT model as an example of how a linear model compares to the best performing ANFIS models. As seen in this table, REr values for QRT are slightly better than FCM in all ranges of ARI, except for $Q_5$, where FCM performs slightly better than QRT. In terms of $Q_{\_Ratio}$, QRT performs better than FCM except for $Q_2$. In terms of RMSE, QRT has a better performance in higher ranges of ARI. In terms of RBias and RMSNE, QRT appears to have better results in the low to mid ranges of ARI. In terms of RRMSE, both FCM and QRT models perform very similarly, with QRT performing slightly better. As shown in Figure 9, boxplots of RE values for FCM and QRT for each quantile are compared. Based on the sizes of the boxes and the number of outliers for each quantile, both models perform very similarly for $Q_2$ and $Q_5$. However, QRT is slightly better for $Q_{10}$, $Q_{20}$, $Q_{50}$, and $Q_{100}$ because the median lines for each quantile are closer to the zero-reference line, and the boxes are slightly smaller for QRT, meaning that the outputs of the QRT model are more consistent and reliable compared to the FCM model.

**Table 4.** Comparison of important statistics between QRT and FCM as the best ANFIS model.

| Quantile | $Q_2$ | | $Q_5$ | | $Q_{10}$ | | $Q_{20}$ | | $Q_{50}$ | | $Q_{100}$ | |
|---|---|---|---|---|---|---|---|---|---|---|---|---|
| Statistics | QRT | FCM | QRT | FCM | QRT | FCM | QRT | FCM | QRT | FCM | QRT | FCM |
| $RE_r$ | 43.10 | 51.43 | 39.10 | 38.18 | 35.58 | 39.96 | 38.88 | 46.90 | 40.01 | 53.23 | 48.12 | 59.80 |
| $Q_{T\_pred}/Q_{T\_obs}$ (Median) | 0.93 | 0.95 | 0.97 | 1.07 | 0.97 | 1.11 | 1.05 | 1.23 | 1.00 | 1.11 | 1.07 | 1.45 |
| RMSE | 49.91 | 50.88 | 115.46 | 119.05 | 190.22 | 206.06 | 289.55 | 315.90 | 507.50 | 531.49 | 749.58 | 845.69 |
| RBias | 31.76 | 44.60 | 32.32 | 67.60 | 38.23 | 87.67 | 41.72 | 53.66 | 46.43 | 44.29 | 57.25 | 61.43 |
| RMSNE | 1.13 | 2.06 | 1.32 | 2.64 | 1.50 | 3.02 | 1.61 | 2.34 | 1.78 | 1.85 | 2.09 | 2.90 |
| RRMSE | 0.82 | 0.84 | 0.75 | 0.78 | 0.77 | 0.84 | 0.79 | 0.87 | 0.88 | 0.93 | 1.01 | 1.08 |

The findings of this study were compared with the results of Aziz et al. [61], in which they applied four different AI-based RFFA methods, ANN, GAANN, GEP, and CANFIS, to estimate design floods using data from 452 small- to medium-sized catchments in the eastern parts of Australia. They reported REr values for the CANFIS method ranging from 34.48 for Q20 to 180.77 for $Q_2$. They also reported median $Q_{\_Ratio}$ values of 0.79 for $Q_{10}$ and 2.81 for $Q_2$ with the best value of 0.95 for $Q_5$. Based on the REr and $Q_{\_Ratio}$ values reported for the CANFIS method, it was found that the results of the FCM method were more reliable, robust, and consistent.

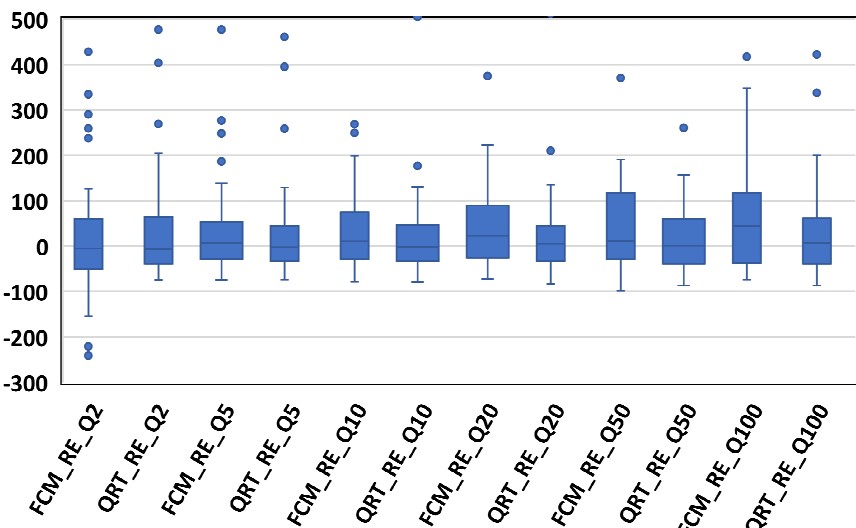

**Figure 9.** Comparison of boxplots of RE values for FCM and QRT.

## 4. Conclusions

This study compares three types of ANFIS models in RFFA, FCM, SC, and GP using data from 181 catchments in south-east Australia (70% being used for training and 30% for testing of the models). FCM has been identified to perform better in terms of REr values. Based on the $Q_{\_Ratio}$ values, SC has performed slightly better than the FCM. The GP has the weakest performance among the three, with notable underestimation of design floods for all the six ARIs. The FCM model provides more accurate flood quantile estimates for NSW catchments than those of Victoria. Also, among all the quantiles, FCM presents the best RFFA model for $Q_5$.

It is recommended to test ANFIS models in different parts of Australia and compare the results with other common linear RFFA models, as this has the potential to become the preferred RFFA method in Australia. The ANFIS model can also be mixed and combined with other methods such as genetic algorithms to reduce model error. Future studies should consider a greater number of predictor variables in model building and testing. Monte Carlo cross validation should also be applied instead of a single split-sample validation. Also, uncertainty analysis should be extended following new approaches [62].

**Supplementary Materials:** The following supporting information can be downloaded at: https://www.mdpi.com/article/10.3390/w14101608/s1, Table S1: Important statistics from the validation (test data) of ANFIS models; Table S2: Important predictor variables for the QRT and ANFIS models.

**Author Contributions:** A.Z. conducted data analysis and drafted the manuscript; A.R. assisted in interpretating results and revised the manuscript; M.V. assisted in running the program, interpreted results, and revised manuscript; B.S. edited the manuscript; and F.A. interpreted the results and edited the manuscript. All authors have read and agreed to the published version of the manuscript.

**Funding:** This research received no external funding.

**Institutional Review Board Statement:** Not applicable.

**Informed Consent Statement:** Not applicable.

**Data Availability Statement:** The datasets used in this study can be obtained from the Australian Bureau of Meteorology, WaterNSW, and Department of Environment, Land, Water and Planning of Victoria by paying a prescribed fee and/or from their websites (free of cost).

**Acknowledgments:** The authors would like to acknowledge the Australian Bureau of Meteorology, WaterNSW, and Department of Environment, Land, Water and Planning of Victoria for providing streamflow data used in this study. This study is part of the first author's PhD research.

**Conflicts of Interest:** The authors declare they have no conflict of interest.

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
