# Peer review of "Regional Flood Frequency Analysis Using the FCM-ANFIS Algorithm: A Case Study in South-Eastern Australia"

_water, doi:10.3390/w14101608_

Round 1
Reviewer 1 Report
The paper is not suitable for publication in the present form. Overall, diverse issues require clarifications:
1-Quantitative performance of ANFIS models should be mentioned for the best combination in the abstract.
2-Motivation and novelty of this research should be mentioned in the introduction.
3-Why did authors use ANFIS models compared to the other robust soft computing tools (i.e., GEP, M5MT, GMDH, EPR, SVM, MARS)?
4-ANFIS model has no novelty at all. Introduction section can be improved by "Ocean Engineering 111, 128-135, 2016", "Journal of Hydrologic Engineering 20 (12), 04015035, 2015".
5-Figure 3 and its descriptions are not suitable for publication.
6-ANFIS results need uncertainty level as done in Water Resources Management 34, 529–561, 2020.
Author Response
The paper is not suitable for publication in the present form. Overall, diverse issues require clarifications:
1-Quantitative performance of ANFIS models should be mentioned for the best combination in the abstract.
Response by authors: The following sentence is added: The FCM performs better in smaller to medium ARIs (2 to 20 years) (ARI of 5 years having the best performance), and in New South Wales State than Victoria.
2-Motivation and novelty of this research should be mentioned in the introduction.
Response by authors: The following sentence is added: “The motivation of this study is to create sound scientific basis on the AI-based RFFA models based on the recent flood data, which will assist upgrade of the RFFA methods in the national guide Australian Rainfall and Runoff.”
3-Why did authors use ANFIS models compared to the other robust soft computing tools (i.e., GEP, M5MT, GMDH, EPR, SVM, MARS)?
Response by authors: The following sentence is added in the last paragraph of the Introduction: “In this study, ANFIS is selected over other AI-based methods as it is one of widely used techniques in hydrology [10, 35].”
4-ANFIS model has no novelty at all. Introduction section can be improved by "Ocean Engineering 111, 128-135, 2016", "Journal of Hydrologic Engineering 20 (12), 04015035, 2015".
Response by authors: Thanks for your suggestions. These two important references are now included in the revised manuscript.
5-Figure 3 and its descriptions are not suitable for publication.
Response by authors: It has been removed in the revised manuscript.
6-ANFIS results need uncertainty level as done in Water Resources Management 34, 529–561, 2020.
Response by authors: Thanks for the comment. In our paper, we have carried out uncertainty analysis by a split-sample validation using several statistics. The approach of the suggested paper will be used in a future study as noted in our conclusion section:
“and uncertainty analysis should also be extended following approaches such as by Saberi-Movahed [62]”.
Reviewer 2 Report
The submitted article (Regional Flood Frequency Analysis using FCM-ANFIS Algorithm - A Case Study for South-eastern Australia) is generally good and fits the journal's scope. However, the following issues must be addressed before further processing:
1- There are grammar and spelling mistakes. Consult an Editor to remove them
2- Avoid pronouns (we, him, she,...etc.) in academic writing. Use a third person format
3. There are some formating mistakes, for example, missed Parentheses in the abstract (38-60%)
4- Avoid mass citations, such as [1, 2-17], [11, 14, 18-23],... . Use one or two references per piece of information
6- Reference the used equations
7- Define all parameters in all equations
8- Number all titles and sub-titles
Author Response
The submitted article (Regional Flood Frequency Analysis using FCM-ANFIS Algorithm - A Case Study for South-eastern Australia) is generally good and fits the journal's scope. However, the following issues must be addressed before further processing:
1- There are grammar and spelling mistakes. Consult an Editor to remove them
Response by authors: The revised manuscript has been edited by a native English Speaker to remove English mistakes.
2- Avoid pronouns (we, him, she,...etc.) in academic writing. Use a third person format.
Response by authors: Pronouns are removed in the revised manuscript and relevant sentences are restructured.
There are some formating mistakes, for example, missed Parentheses in the abstract (38-60%)
Response by authors: All the formatting mistakes are removed.
4- Avoid mass citations, such as [1, 2-17], [11, 14, 18-23],... . Use one or two references per piece of information
Response by authors: Thanks for the suggestions. This is fixed in the revision manuscript by adding only one or two relevant references.
6- Reference the used equations
Response by authors: All the equations are referenced/quoted in the body of the paper.
7- Define all parameters in all equations
Response by authors: All the parameters in the equations are defined in the revised manuscript.
8- Number all titles and sub-titles
Response by authors: All the titles and sub-titles are numbered.
Round 2
Reviewer 1 Report
Accept as is